# High-speed blind structured illumination microscopy via unsupervised algorithm unrolling

Zachary Burns [1,8], Junxiang Zhao[1,8], Ayse Z. Sahan [2,3], Jin Zhang [2,4,5] & Zhaowei Liu [1,6,7]

Blind structured illumination microscopy (blind-SIM) is a valuable tool for achieving super-resolution without the need for known illumination patterns. However, in its current formulation the algorithm requires many iterations to converge, leading to long inference times and limited use for real-time or video-rate imaging. We present unrolled blind-SIM (UBSIM), an algorithm which integrates a learnable neural network inside the unrolled iterations of the blind-SIM algorithm. UBSIM delivers a reconstruction speed two to three orders of magnitude faster than that of current iterative blind-SIM methods, while achieving similar resolution and image quality. Furthermore, we demonstrate that UBSIM can be trained in an unsupervised manner that reduces hallucinations and produces superior generalization capability when compared to benchmark super-resolution networks. We test UBSIM experimentally on live cells and present video-rate super-resolution imaging up to 50 Hz. Using our method, we observe dynamic remodeling of the endoplasmic reticulum with high spatiotemporal resolution.

Structured illumination microscopy (SIM) is a widely used super-resolution (SR) technique that illuminates a sample with spatially non-uniform light to recover high-resolution information[1,2]. While SIM has a moderate two-fold resolution improvement when compared to other SR techniques such as STORM or STED, it requires only a handful of raw images to generate an SR frame, and thus provides high temporal resolution from both the data acquisition perspective and the SR reconstruction perspective. Over the past decade, SIM has been demonstrated as an attractive SR method for cases where video-rate imaging is needed or when exposure time is limited due to photobleaching[3–5].

Various methods of SIM have been demonstrated using either periodic or random illumination patterns[6,7]. Standard periodic illumination SIM requires precise knowledge of the illumination patterns and their relative locations to correctly retrieve the frequency domain information of the object; thus, it is highly sensitive to these parameters. Any mismatch between the assumed and actual illumination patterns leads to artifacts and degraded performance[8]. Therefore, standard SIM systems need to be carefully calibrated and maintained for various imaging conditions[9], which limits their practical applications in biological research.

To overcome this limitation, the blind-SIM algorithm was first developed by Mudry and colleagues to utilize randomized illumination patterns for SR image reconstruction[10]. As an iterative, model-based reconstruction method, blind-SIM solves jointly for the object and illumination patterns based on constraints of illumination statistics.

[1]Department of Electrical and Computer Engineering, University of California, San Diego, 9500 Gilman Drive, La Jolla, CA, USA. [2]Department of Pharmacology, University of California, San Diego, 9500 Gilman Drive, La Jolla, CA, USA. [3]Biomedical Sciences Graduate Program, University of California, San Diego, 9500 Gilman Drive, La Jolla, CA, USA. [4]Department of Bioengineering, University of California, San Diego, 9500 Gilman Drive, La Jolla, CA, USA. [5]Department of Chemistry & Biochemistry, University of California, San Diego, 9500 Gilman Drive, La Jolla, CA, USA. [6]Material Science and Engineering Program, University of California, San Diego, 9500 Gilman Drive, La Jolla, CA, USA. [7]Center for Memory and Recording Research, University of California, San Diego, 9500 Gilman Drive, La Jolla, CA, USA. [8]These authors contributed equally: Zachary Burns, Junxiang Zhao. ✉e-mail: zhaowei@ucsd.edu

Several other iterative blind-SIM algorithms have since been developed based on various constraints[11–14]. A thorough comparison of our work with the existing blind-SIM literature is provided in the supplementary material (Supp. Table 1). Blind-SIM has proven to be a powerful technique that has allowed the design of SIM systems which rely on random illumination patterns that do not require calibration and are easier and cheaper to build and maintain.

While blind-SIM has greatly reduced the hardware constraints of SIM, it comes with the tradeoff of a slow, iterative reconstruction process. Even with GPU-based hardware acceleration, the time for a single reconstructed frame can range from tens of seconds to minutes, depending on the image size and number of input frames. Thus, reconstruction of videos becomes prohibitively long, and real-time reconstruction is challenging. Ideally, users of an SR system should be able to rapidly reconstruct a high-resolution image on the fly in order to evaluate a sample immediately. Widespread adoption and use of SR technologies by target groups, such as the biology community, relies on the ease of use of such systems.

Recently, data-driven learning-based techniques such as deep learning have become widely used for SR image reconstruction. These methods often have much faster inference times than traditional iterative reconstruction algorithms and can produce higher quality results in challenging conditions, such as shorter exposure times with lower signal-to-noise ratios[15–18]. However, there are several critical limitations to deep neural networks (DNNs) that prevent their widespread use for SR microscopy.

Firstly, most methods using DNNs for SR rely on supervised training strategies where datasets of matched low and high-resolution images are needed. Generating a training set of experimental, paired images is not typically feasible; therefore, simulated data is generated using various types of objects combined with the physical model of the imaging process. Yet, it is often unclear how well these models will perform when applied to an experimental system where the types of objects may differ from the objects the model is trained on. It is well documented that DNNs can overfit to training data and hallucinate when applied to new, unseen data[19–21]. This lack of generalizability is a major concern with the use of DNNs in biological imaging.

Secondly, neural networks are so-called black box systems and are not readily interpretable. Unlike iterative algorithms, when trained in an end-to-end manner, DNNs typically do not explicitly incorporate any knowledge of the physical imaging model[22]. Thus, it is hard to determine whether a DNN is truly learning the more general inverse process of the forward model or simply relying on image statistics from the training set to predict the most likely object features.

There has been growing interest in combining physics with DNNs for image reconstruction[23–26]. One promising technique, known as algorithm unrolling (or unfolding), uses trainable parameters within the framework of iterative algorithms[27–30]. In algorithm unrolling, each iteration of an iterative algorithm becomes a layer in a DNN, and multiple layers are connected end-to-end, forming the network. In this way, the number of layers in the network is equivalent to running an iterative algorithm for the same number of iterations. Within each layer, learnable parameters can be included, and the entire network can be trained via backpropagation to update these parameters. Unrolled algorithms have been shown to be interpretable, generalizable, and require fewer parameters and training data than traditional DNNs.

In this paper, we propose a new unrolled algorithm we name unrolled blind-SIM (UBSIM). UBSIM produces a two-fold improvement in resolution with a reconstruction speed that is two to three orders of magnitude faster than iterative blind-SIM in both simulations and experiments. Furthermore, we demonstrate that UBSIM generalizes to unseen data far better than reference SR DNNs. Lastly, to illustrate its applicability to biological research, we perform live-cell imaging of endoplasmic reticulum dynamics with a high-speed SIM acquisition setup utilizing a synchronized DMD illumination system. Using UBSIM, we capture the real-time movement and collapse of endoplasmic reticulum network structures.

## Results

### Derivation of unrolled algorithm

To begin, we describe the unrolled blind-SIM algorithm following the derivation of the iterative algorithm developed by Mudry and colleagues in their paper[10]. We adopted the same notation for consistency.

For structured illumination microscopy, the forward model for each sub-frame M is:

$$M = (I\rho)*h \tag{1}$$

where $I$ is the illumination pattern, $\rho$ is the object and $h$ is the point spread function of the detection objective. The aim of SR imaging is to recover $\rho$ given the set of diffraction-limited images M. In traditional SIM the illumination patterns $I$ are known and only $\rho$ is solved for. However, in blind-SIM, we must jointly solve for $\rho$ and $I$ as the illumination patterns are unknown. This is achieved by making an additional assumption that all the illumination patterns on average sum to a uniform field:

$$\sum_{l=1}^{L} I_l \approx L I_0 \tag{2}$$

Where $L$ is the number of sub-frames and $I_0$ is a constant equal to the average intensity received by the sample. The error in the uniform approximation drops at a rate of $\frac{1}{\sqrt{L}}$. See Supplementary Section S3 for derivation.

To ensure positivity, $I_l$ and $\rho$ are rewritten as the square of auxiliary variables such that:

$$I_l = i_l^2$$
$$\rho = \xi^2 \tag{3}$$

The cost function to evaluate potential solutions for the object and illumination patterns can be written as:

$$F(\xi, i_1, \ldots, i_{L-1}) = \sum_{l=1}^{L-1} \|M_l - (\xi^2 i_l^2)*h\|^2 + \|M_L - \xi^2 I_L*h\|^2 \tag{4}$$

with:

$$I_L = L I_0 - \sum_{l=1}^{L-1} i_l^2 \tag{5}$$

To solve for the object and illumination patterns, the gradients of the cost function with respect to the object and illumination patterns are calculated, and the solutions are updated by a weighted step:

$$\xi_n = \xi_{n-1} + \alpha_n g_{n,\xi}$$
$$i_{l,n} = i_{l,n-1} + \alpha_n g_{n,i} \tag{6}$$

where $\alpha$ is the step size, $g$ is the gradient, and $n$ is the iteration number. This update constitutes one iteration of standard gradient descent and is repeated for either a fixed number of iterations or until the reduction in the cost function plateaus. The Polak-Ribière conjugate gradient method[31], used by Mudry and colleagues, as well as Nesterov acceleration[32], can also be used to significantly accelerate optimization. Details of the implementation of these methods are included in Supplementary Section S4.

Inspired by recent work in algorithm unrolling, we propose using a trainable neural network which operates on the gradient as follows

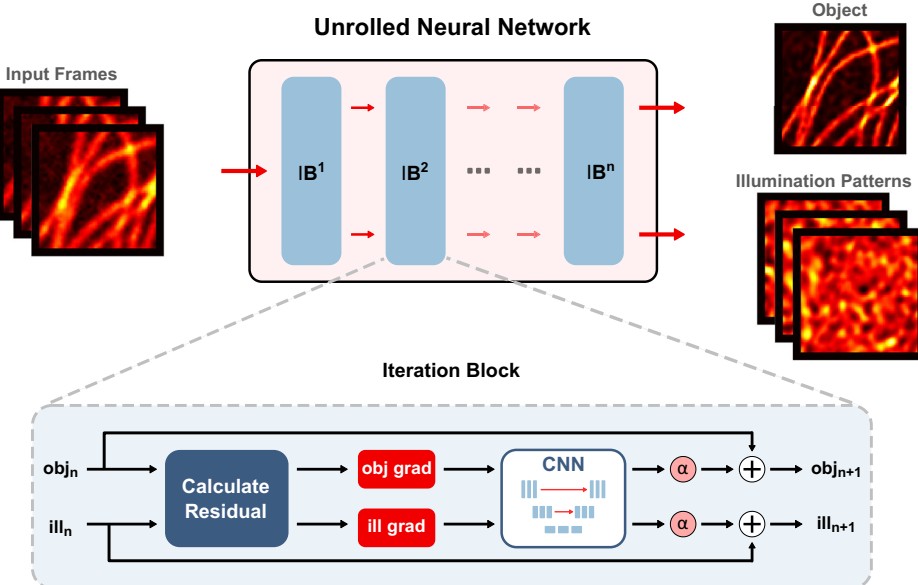

**Fig. 1 | Schematic diagram of the unrolled blind-SIM algorithm.** The network consists of several iteration blocks (IBs) which mimic the update steps of a traditional iterative algorithm. The inset shows the update equations applied in each iteration block to the object (obj) and illumination patterns (ill). Grad is the gradient with respect to each, $\alpha$ is the step size, and CNN is a convolutional neural network. n indicates the iteration step.

for each iteration:

$$\xi_n = \xi_{n-1} + \alpha_n \mathrm{CNN}(g_{n,\xi})$$
$$i_{l,n} = i_{l,n-1} + \alpha_n \mathrm{CNN}(g_{n,i,l}) \tag{7}$$

where CNN is a convolutional neural network for which the weights are learned to minimize the cost function across a set of training data. For our algorithm, only one set of weights was learned for one CNN, which was applied recursively at each iteration.

Figure 1 shows a block diagram for UBSIM. The network consists of a fixed number of iteration blocks, which serve to update the object and illumination patterns in a similar manner to a traditional iterative algorithm. In each iteration block, the residual from the cost function is used to calculate the gradients for the object and each illumination pattern. The gradients (stacked as channels in a tensor object) are then used as inputs to a CNN. The CNN's scaled outputs are added to the original inputs to obtain the updated object and illumination patterns. The CNN operates directly on the gradients similar to how a preconditioner[33,34] or update direction calculation[31,32,35] is used to accelerate convergence for iterative gradient descent algorithms. In this way, the use of the network can be thought of as a type of learned preconditioner[36,37] or learned gradient descent[38-41] for which there are examples from literature.

We employed an unsupervised training process to optimize the weights of the CNN in the unrolled neural network. We used the cost function loss of equation 4 to update the CNN weights during the training step rather than a loss function directly comparing the output object and ground truth object. This has the added benefit of removing the need for ground truth training data and helps with generalization.

For the unrolled network, we selected a modified U-Net architecture with 2 rescaling layers and skip connections[42] (Supplementary Section S5). We chose this architecture because it is widely used for many image processing tasks and has been shown to perform well in denoising. Furthermore, with less than 250,000 parameters, this model is still relatively small compared to many state-of-the-art super-resolution models. More details on the training process can be found in the methods section and supplementary material.

## Evaluation on simulated data

To evaluate the performance of UBSIM, we first tested the algorithm on a set of simulated fluorescence images that were generated from experimental microscopy data[43] (see supplementary material, S7). Simulated data provides ground truth images which we can use to quantitatively and qualitatively evaluate the model's performance.

In Fig. 2a we quantitatively compared the image resolution improvement using several standard image quality metrics on a set of test data which were not included in the training process. The mean-squared error (MSE), structural similarity index measure (SSIM) and peak signal-to-noise ratio (PSNR) all showed significant improvement for the UBSIM results compared to the diffraction-limited widefield images, indicating an improvement in image fidelity. To estimate the improvement in spatial resolution, we employed the widely used image decorrelation analysis method[44]. The UBSIM results on average showed a resolution improvement of approximately 2-fold, which matches the expectation, as the simulated illumination speckles were also diffraction-limited with the same numerical aperture (NA) as the detection objective.

Comparisons of reconstruction results on four types of cellular structures are shown in Fig. 2b. The cell structures used were microtubules (MT), clathrin-coated pits (CCP), endoplasmic reticulum (ER), and F-actin filaments. The results from UBSIM revealed structural features that were not visible from the widefield images and matched well with the ground truth images. For reference, the results from iterative blind-SIM are also displayed. While iterative blind-SIM provided a similar level of resolution improvement, the main advantage of UBSIM is the reduced inference time and greater resistance to noise overfitting (see Supplementary Section S8). Specific examples of structural feature recovery are presented in Fig. 2c, d. Two regions from the microtubule and CCP examples in Fig. 2b are displayed in Fig. 2d. UBSIM was able to reveal closely spaced MT filaments and the ring-like structures in CCPs. Line profiles taken from both regions in Fig. 2c show close alignment with the ground truth and demonstrate the ability of UBSIM to resolve sub-diffraction features.

## Comparison of inference time

In Fig. 3 we quantitatively compared the speed for unrolled and iterative blind-SIM. For iterative blind-SIM, we ran evaluations using

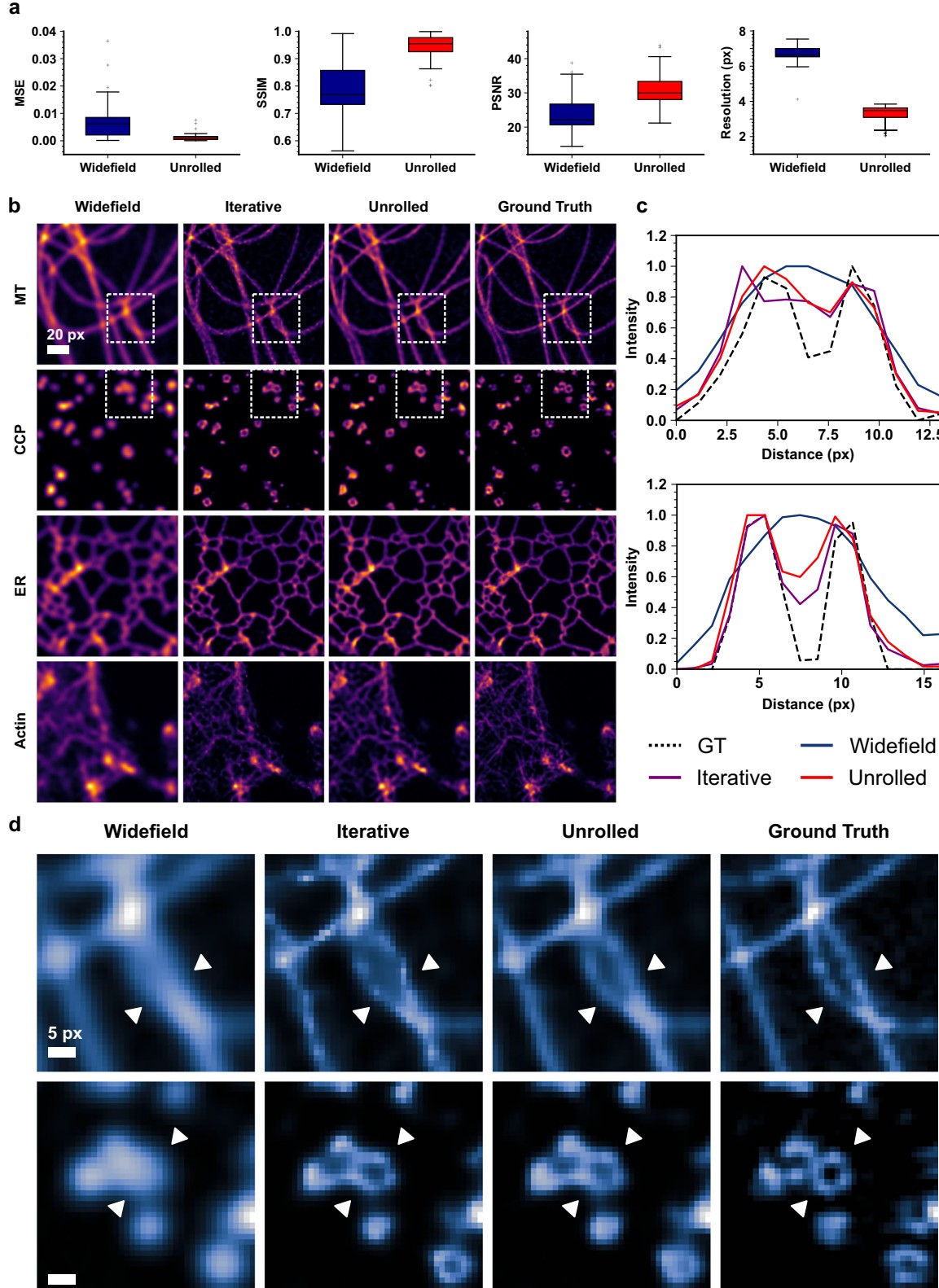

**Fig. 2 | Unrolled blind-SIM performance on simulated data. a** Quantitative comparison of diffraction-limited images and UBSIM outputs with ground truth objects on a series of image quality metrics. Boxplots are generated using $n = 100$ independent samples where each sample is a simulated test image not included in the training or validation datasets. The center line indicates the median, the box edges denote the first and third quartiles, and the whiskers extend to the maximum and minimum datapoints within 1.5x the interquartile range. Outlier points are plotted beyond the whiskers. **b** Comparison of iterative and unrolled blind-SIM with ground truth and widefield (WF) images on different cellular structures. Scale bar: 20 pixels. **c** Line profile comparison of indicated regions from the cropped regions in (**d**). **d** Selected cropped regions from the images in (**b**). Scale bar: 5 pixels.

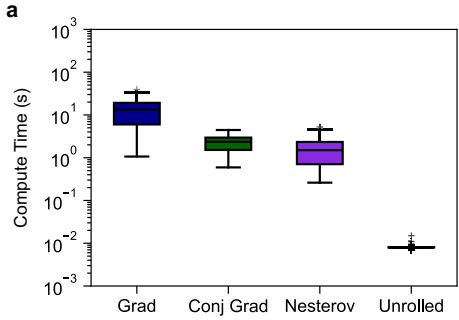
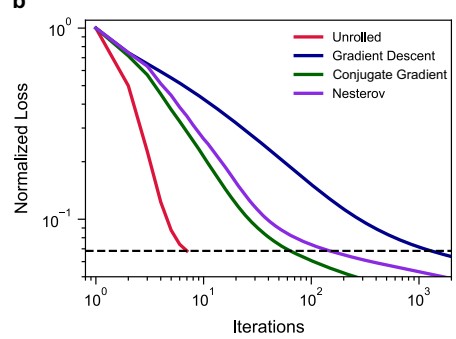

**Fig. 3 | Inference time comparison of iterative methods versus unrolled blind-SIM. a** Comparison of computation time for image reconstruction. Boxplots are generated using $n = 100$ independent samples where each sample is a simulated test image not included in the training or validation datasets. The center line indicates the median, the box edges denote the first and third quartiles, and the whiskers extend to the maximum and minimum datapoints within 1.5x the inter-quartile range. Outlier points are plotted beyond the whiskers. **b** Comparison of normalized loss versus iteration number. Lines are the average values for 100 test set images. Dashed black line marks the average normalized loss for the unrolled network after 6 iterations.

regular gradient descent as well as two accelerated methods: nonlinear conjugate gradient descent[31] and gradient descent with Nesterov momentum[32]. Additional information on the implementation of these methods is included in Supplementary Section S4. There are two ways we assessed the speed: (1) the number of iterations to reach a target loss and (2) the time taken for the computation. For each image in the test dataset, we first calculated the final cost function value for the result from UBSIM. For this paper we chose a UBSIM model with six iteration blocks (see Supplementary Section S10). Next, we ran the iterative blind-SIM methods until their cost functions reached the same level as the UBSIM result and recorded the time and number of iterations. For a fair comparison, all algorithms were written in PyTorch and run on the same GPU (Nvidia RTX 1080 Ti).

We found that UBSIM was on average 3 orders of magnitude faster than the gradient descent implementation and 2 orders of magnitude faster than the conjugate gradient descent and Nesterov acceleration methods in terms of computation time. UBSIM required 2 orders of magnitude fewer iterations than gradient descent and 1 order of magnitude fewer than the accelerated methods. A single $256 \times 256$ pixel image took approximately 10-20 seconds to reconstruct for gradient descent, 1-3 seconds for the accelerated methods, and 10 milliseconds for UBSIM. The UBSIM computation time improvement was relatively faster than the iteration improvement due to differences in how PyTorch implements the two functions. UBSIM has a fixed number of iterations, and thus PyTorch uses a static graph, whereas iterative blind-SIM has a variable iteration number and requires a dynamic graph. While absolute computation time will vary with hardware, the GPU used here is a consumer-grade card four generations older than current, and thus is of negligible cost for many labs. With UBSIM, real-time image reconstruction for video-rate blind-SIM is possible. In this work, we used a raw image framerate of 100 fps for our high-speed imaging demonstration. The highest speed experimental data in this paper produced SR images at 50 frames per second, which required a 20 ms capture time. Therefore, even with modest computation power, real-time SR video reconstruction is possible. The improvement in speed greatly increases the practical usability of blind-SIM for potential users such as biologists, as they can now quickly view SR imaging results rather than waiting hours for reconstruction of a single video.

## Evaluation of generalization performance

We tested the generalization capability of UBSIM and three common SR networks by training them on three separate datasets and cross-comparing their performance on each. The SR networks we chose to compare with are the U-Net[42], Enhanced Deep Residual Network (EDSR)[45], and Residual Channel Attention Network (RCAN)[46]. All of these networks have been widely used for SR image reconstruction. We generated three simulated datasets consisting of ER, CCPs, and digits from the MNIST[47] dataset. Each of these objects has distinct structures which are different from each other. The U-Net, EDSR, and RCAN networks were all trained in a supervised manner where the MSE loss between the predicted and ground truth data was used to update the network weights. UBSIM was trained in the same unsupervised method as detailed earlier. The networks were trained on the ER, CCP, and MNIST datasets separately, and then each trained model was applied to test sets from each object type.

A qualitative comparison of the networks' generalization ability is shown in Fig. 4a. In this figure, we compared the output of each trained model when applied to a single example from the ER test dataset. The top row shows results for each network type trained on ER data. As expected, when the training and test data were drawn from the same distribution, all networks produced images that show increased resolution and matched well with the ground truth image. However, when the models were tested on out-of-distribution data, a clear difference emerged between the unrolled network and the standard models. The middle and bottom rows display the model outputs on the ER test data when trained on CCP and MNIST data, respectively. UBSIM produced results which were visually similar to the results produced from the ER-trained model and again showed high fidelity to the ground truth image. Conversely, the results from U-Net, EDSR, and RCAN all showed signs of hallucination-based artifacts. For example, when trained on CCP data, the standard models produced results that have dot-like characteristics that are similar to those seen in the CCP data. When trained on the MNIST-based dataset, the models again showed artifacts that are characteristic of the dataset, appearing overly saturated with uniform object intensity. Cropped regions from these comparisons are displayed in Fig. 4b.

We quantitatively compared UBSIM to standard networks using the Learned Perceptual Image Patch Similarity (LPIPS)[48] metric for each network and combination of training and test datasets. The LPIPS metric is a perceptual image metric that compares the differences in the extracted features of images from a pretrained convolutional neural network (VGG[49] in our case) and has been shown to match human perception well. We chose a perceptual metric as it is highly sensitive to changes in object structure, indicating if artifacts are occurring.

Figure 4c contains a matrix for each network architecture with every combination of training and test data. Diagonal elements of the matrix represent cases where the training and test data types are the same, whereas all the other elements represent cases where they

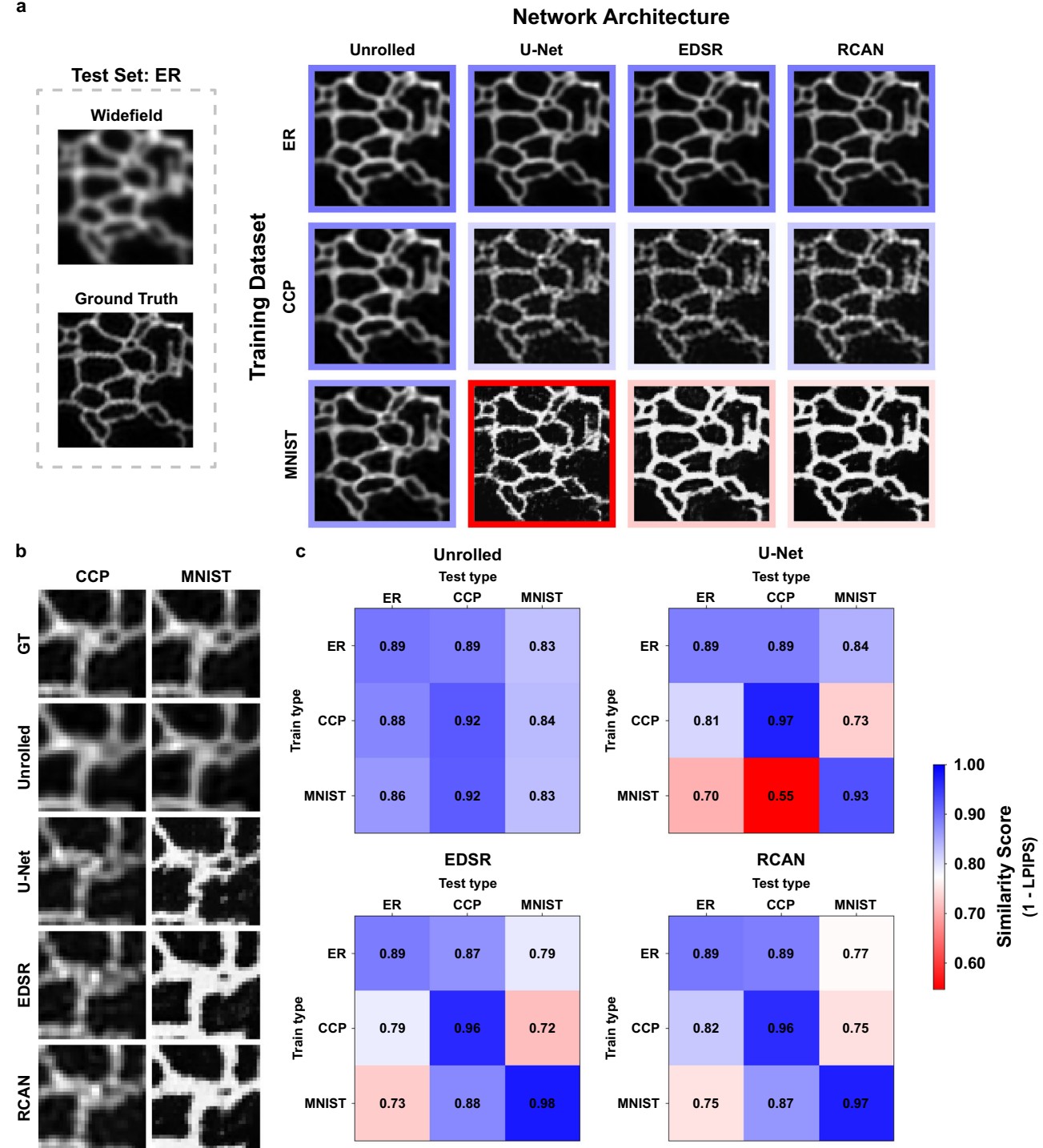

**Fig. 4 | Comparison of generalization performance of various SR network architectures. a** Predictions from various SR models trained on out-of-distribution datasets tested on an ER sample. Rows represent different training datasets, and columns represent different model architectures. Border color indicates similarity score value. **b** Comparison of the cropped region from a. GT is the ground truth image. **c** Quantitative comparison of models when tested on cross-domain data using an LPIPs-based similarity metric.

differ. Each element in the matrix contains the average LPIPS-based similarity score for 100 images in a test dataset. Here, we define the similarity score as 1 − LPIPS, so that the value of 1 indicates perfect similarity.

Generalization performance can be determined by assessing the change in similarity score across each column of the matrix. The figure shows that UBSIM maintains relatively consistent similarity scores across the different test object data types, while the other SR networks exhibited larger changes when applied to new objects.

## Live-cell imaging of actin

Next, we demonstrated the ability of UBSIM to produce SR images on experimental live-cell data. We employed an automated system to control the projection of random, diffraction-limited speckle

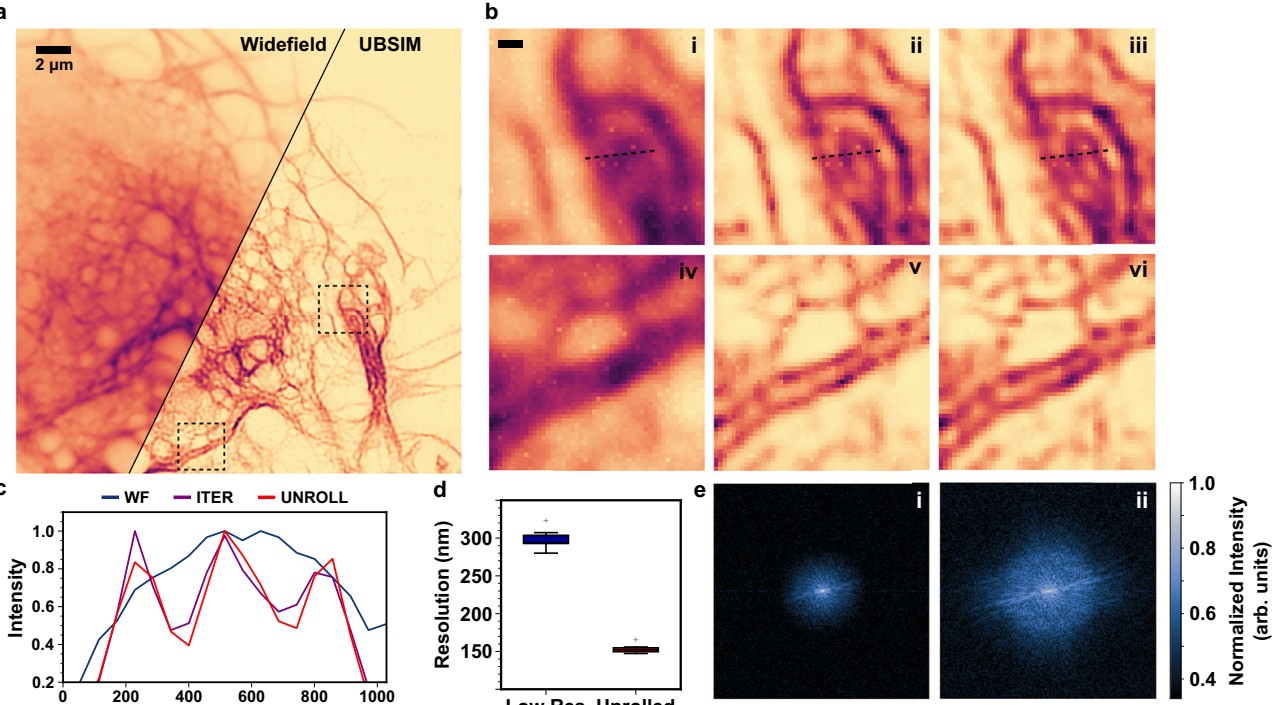

**Fig. 5 | UBSIM imaging of actin-stained live COS-7 cells. a** Overlay image comparing widefield (top left) and UBSIM (bottom right) images of a live COS-7 cell transfected with Lifeact-mNeonGreen to visualize actin. **b** Cropped areas from (**a**) with conditions: (i, iv) widefield (ii, v) iterative (iii, vi) unrolled. Scale bar: 300 nm. **c** Line profiles are taken from the areas indicated in (i-iii). **d** Calculated resolution of widefield and UBSIM images. Boxplots are generated using $n = 10$ repeated measurements from the same sample. The center line indicates the median, the box edges denote the first and third quartiles, and the whiskers extend to the maximum and minimum datapoints within 1.5x the interquartile range. Outlier points are plotted beyond the whiskers. **e** Comparison of Fourier-space of widefield (i) and UBSIM (ii) images.

illumination patterns onto our samples using a digital micromirror device (DMD). The DMD was synchronized with the camera to ensure that the patterns did not move and were not averaged out during each image exposure period. More details of the experimental setup can be found in Supplementary Section S11.

We first imaged live COS-7 cells, fibroblast-like cells derived from monkey kidney tissue, expressing mNeonGreen-tagged Lifeact, a short peptide which binds F-actin[50]. The results are shown in Fig. 5. The UBSIM model was trained with the same simulated dataset as Fig. 2 and contained an even mix of the 4 cell structure types.

UBSIM resulted in a 1.94x resolution improvement over the widefield images, as confirmed by the image decorrelation metric. The estimated resolution dropped from approximately 300 nm to 150 nm after the unrolled network. Furthermore, resolution improvement was indicated by increased high-spatial-frequency content in the normalized logarithmic Fourier magnitude spectra of the widefield and UBSIM images. We were able to resolve closely spaced actin filaments that could not be seen in the widefield images (Fig. 5b, c). Additionally, the results from UBSIM matched well with the results obtained from iterative blind-SIM, indicating that hallucinations were not occurring due to the mismatch between the simulated data used to train the model and the experimental data on which it was tested. As iterative blind-SIM has no learned components, it can serve as a good unbiased benchmark to test against when there is sufficient signal-to-noise ratio.

### Video imaging of ER dynamics

To demonstrate video-rate imaging using UBSIM, we imaged the spatiotemporal dynamics of ER remodeling in real-time. The ER regulates protein synthesis, folding, and modifications[51] and is a highly dynamic organelle which forms a dense network with multiple structural elements, including cisternae (sheet-like structures), and tubules[52,53]. The dense network and rapid movements of the ER make it a particularly challenging organelle to image[54].

To demonstrate the ability of UBSIM to capture the dynamic movements of the ER in live cells, we first used COS-7 cells expressing ER-stagRFP[55], a red fluorescent protein targeted to the cytoplasmic side of the ER via a transmembrane signal peptide[56]. Figure 6a–e presents the UBSIM imaging results. We modulated the illumination patterns at 20 fps and used 20 frames for reconstruction. To visualize dynamics, we used a rolling-frame reconstruction method with a 2-frame step to achieve an effective reconstructed video speed of 10 fps. Because this method includes a large overlap in frames between reconstructions, it introduces temporal smoothing, but can be appropriate for ER where the dynamics of interest (compartment remodeling) occur on timescales of seconds and are generally slower than the total SR frame integration time[57,58]. Figure 6e, shows the dynamics of a tubule region in the ER which could not be seen from the widefield images (Fig. 6d). ER tubules are generally peripherally located and exhibit membrane curvature as opposed to the sheet-like cisternae structures[52]. Over the course of approximately two seconds, the tubule collapsed as the ER network began to remodel and form a new tubule structure. Additional videos of the changing ER networks are included as supplementary material (Supplementary Movies 1–5).

Figure 6f–j presents another set of the UBSIM imaging results for ER, this time expressing ER-mNeonGreen (a green fluorescent protein). The raw images were captured at 100 fps. Using the same rolling window and number of input frames the reconstructed SR video is at 50 fps. Example video reconstructions are included in the supplementary material. We observed the rapid collapse of a tubule structure in just under one second as shown in Fig. 6i, j. ER remodeling, like what we observed here in our test cases, can function to maintain homeostasis and is dynamically regulated in response to changing cellular

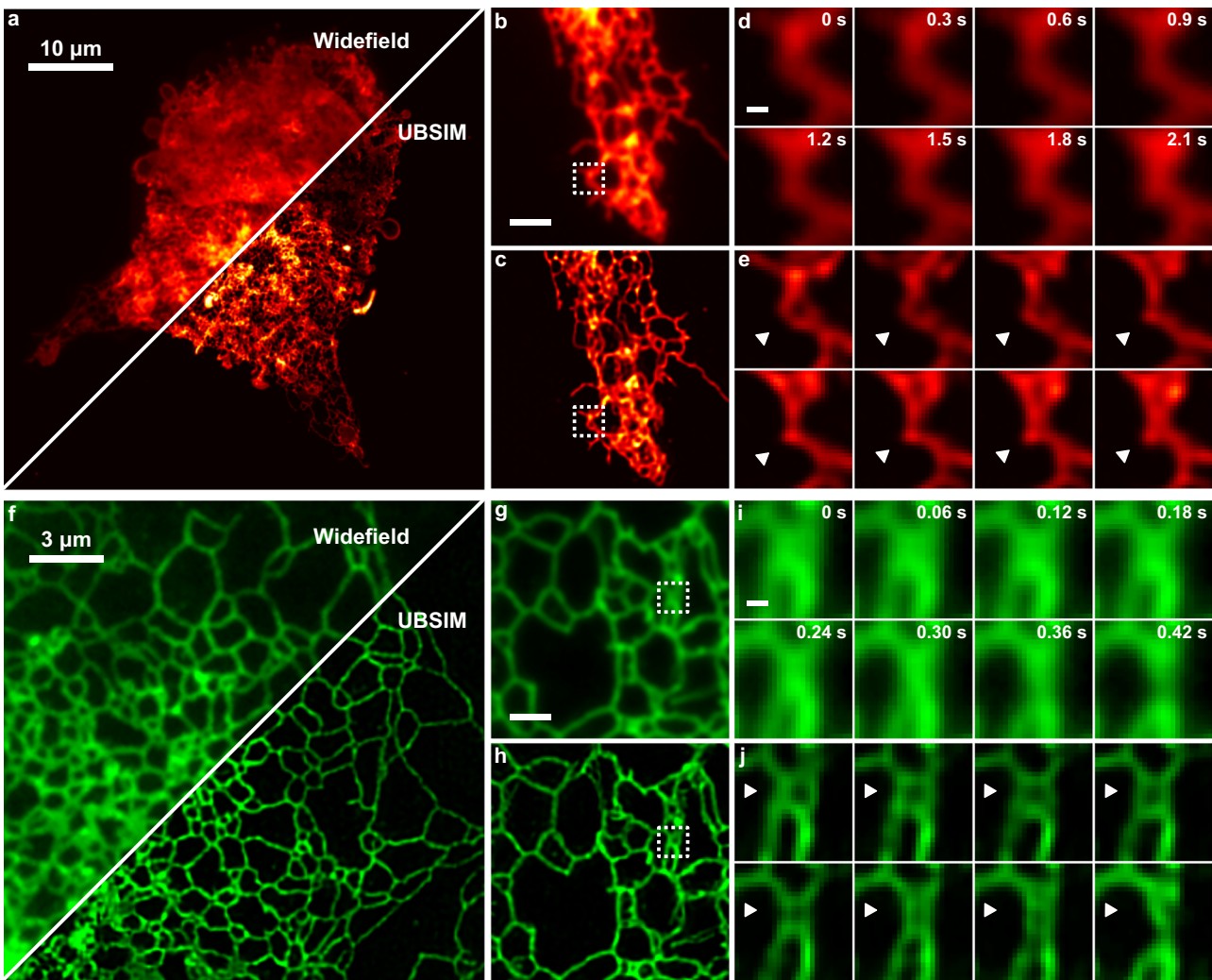

**Fig. 6 | Videos of ER dynamics. a** Widefield (top left) and UBSIM (bottom right) images of a COS-7 cell transfected with ER-stagRFP. **b, c** Comparison of widefield (**b**) and UBSIM (**c**) images from a dynamic ER region in a COS-7 cell. Scale bar: 2 μm. **d, e** Timelapse of widefield (**d**) and UBSIM (**e**) images from the cropped region in (**b**, **c**). Scale bar: 300 nm. **f** Widefield (top left) and UBSIM (bottom right) images of a COS-7 cell transfected with ER-mNeonGreen. **g, h** Comparison of widefield (**g**) and UBSIM (**h**) images on the dynamic ER region in a COS-7 cell. Scale bar: 1.5 μm. **i, j** Timelapse of widefield (**i**) and UBSIM (**j**) images from the cropped region in (**g**, **h**). Scale bar: 250 nm. In this figure, four separate cells are imaged. One each for: **a,b–e**, **f**, and **g–j**. Two additional unique cells are imaged in Supplementary Movies 4 and 5.

conditions or stress[59]. Our approach can thus capture the complex and rapid remodeling of the ER and can be utilized in the future to characterize ER dynamics under various stressed and diseased conditions.

## Discussion

We present the unrolled blind-SIM algorithm (UBSIM), which can reconstruct SR images from unknown illumination patterns with much faster speeds than current methods. By incorporating learnable parameters with the unrolled, physics-based iterations of the blind-SIM algorithm, we can achieve high-resolution images with many fewer iterations, thus reducing the inference time by orders of magnitude. UBSIM combines the benefits of neural networks (fast inference time) and iterative algorithms (no data bias, physics-based interpretability). Using simulated data, we verified the fidelity of UBSIM results against ground truth images and quantified the speed improvement. Additionally, we studied the effect of training data composition on image reconstruction and found that UBSIM generalizes better to unseen data than benchmark SR networks. We further demonstrated live-cell imaging with UBSIM and video-rate imaging with up to a 50 Hz reconstructed frame rate.

Our current demonstration results in a 2× improvement in resolution, as we used diffraction-limited speckle patterns for illumination. However, this improvement is fundamentally limited by the resolution of the illumination patterns, not by the UBSIM algorithm. In future work, we aim to combine high-index substrates with UBSIM to further improve the resolution below 100 nm.

While UBSIM improves on current blind-SIM algorithms, several limitations must be considered by potential users. Firstly, UBSIM requires training before it can be used for inference, which creates an upfront time cost. For one-off reconstructions, iterative methods, despite their longer reconstruction times, may be preferred. However, for cases that include repeated image reconstruction, such as video imaging or repeated experiments, UBSIM can provide a significant time advantage. Secondly, UBSIM, like all blind-SIM methods, requires multiple sub-frames for each reconstructed SR frame, which can affect image acquisition speeds. In our work we utilize a sliding-frame reconstruction approach in addition to our high-speed illumination system to boost acquisition speed. However, this approach can cause temporal smoothing and is not suitable for cases where the sample movement is generally faster than the effective integration time for a

single SR frame. Further improvements in blind-SIM acquisition speed require either a reduction in sub-frames per reconstructed image or increased speckle modulation and camera capture rates.

We envision UBSIM as a useful tool for those in the SR community interested in low-cost, flexible high-speed imaging. UBSIM can enable real-time, video-rate super-resolution with modest hardware requirements. Additionally, as UBSIM does not require knowledge of illuminations patterns, users can produce SR video without the need for calibrated patterns. Lastly, UBSIM's superior generalization performance allows users to have confidence in the fidelity of reconstructed images without having to carefully consider whether their training dataset matches the test data. Going forward, UBSIM can be harnessed to study the spatiotemporal dynamics of subcellular organelles, like the ER, and uncover the complex biology of the cell.

## Methods

### Algorithm code and training

The UBSIM network and iterative blind-SIM codes were written in PyTorch 2.1.2 and Python 3.10.14. A simulated dataset was generated using images from the BioSR dataset[43] and the physical forward model for speckle-based incoherent structured illumination. The simulated data was generated using code written in MATLAB release R2021b. Code and the datasets are provided for open-source use on GitHub and Zenodo, respectively (UBSIM code is available at https://github.com/Zach-T-Burns/Unrolled-blind-SIM, datasets and simulation code are available at https://zenodo.org/records/17852915). Models were trained on an Nvidia A6000 Ada GPU, and inference testing was performed on an Nvidia RTX 1080 Ti GPU. For a typical training run, models used an initial learning rate of 1e-4 with a reduction upon plateauing and were trained for 200 epochs, taking about 2 h. Specific details on datasets, physical model parameters, and training details for each trained model instance can be found in the supplementary material (Section S6).

### Optical setup

Live-cell imaging was conducted on a customized Olympus IX83 inverted microscope. A 488 nm CW laser (Coherent Genesis MX488–1000 STM) and a 532 nm CW laser (Laser Quantum Ventus 532) were combined and coupled into a multimode fiber (Thorlabs, core diameter: 50 μm, NA 0.22). The fiber-coupled laser was then collimated and projected to a DMD (Texas Instruments DLP4710LC), which generates random illumination patterns. The random patterns were then projected onto the sample plane using a tube lens and an objective lens (Olympus 60X, 0.8 NA) to produce diffraction-limited speckle illumination. The fluorescence signal was detected with a water immersion objective (Olympus 60X, 1.2 NA, effective 1.0 NA in thick sample) and bandpass filters (Chroma ET520/40 m for mNeon-Green, Semrock FF01-588/21-25 for stag-RFP). The detected images were captured with a sCMOS camera (Hamamatsu ORCA Flash 4.0 V3), which was synchronized with the DMD so that each frame is captured with a stabilized speckle illumination pattern. The software HCImage version 4.4.5 was used for image collection.

### Cell growth and preparation

COS-7 cells used in this study were acquired from ATCC (CRL−1651). Cells were cultured in Dulbecco's modified Eagle medium (DMEM; Gibco 11995-065) including 4.5 g/L glucose, 4 mM L-glutamine, 1.5 g/L sodium bicarbonate, and 1 mM sodium pyruvate, which was supplemented with 10% fetal bovine serum (FBS, Gibco 26140-079), and 1% (v/v) penicillin/streptomycin (Pen/Strep, Gibco 15140−122). Mycoplasma contamination was tested routinely in cells and found to be negative. For experiments, cells were plated onto glass slides two days prior to imaging. 16-24 hours after initial seeding at ~70% confluency, cells were transfected with ER-mNeonGreen, ER-stagRFP, or Lifeact-mNeonGreen using Polyjet In Vitro DNA Transfection Reagent (SL100688, SignaGen Laboratories). Cells were incubated overnight, then the transfection medium was replaced with modified Hank's balanced salt solution, including 1x HBSS (diluted from 10X HBSS; Gibco, 14065) with 2g/L glucose at pH 7.4 for imaging.

### Constructs

LifeAct-mNeonGreen was a gift from Dorus Gadella[60] (Addgene plasmid #98877; http://n2t.net/addgene:98877; RRID:Addgene_98877). ER-stagRFP was generated as previously reported[55]. ER-mNeonGreen was generated by restriction enzyme cloning. mNeonGreen was subcloned into the ER-stagRFP construct, replacing stagRFP, using the BamHI and XbaI restriction sites. All constructs were verified by forward and reverse sequencing.

### Imaging conditions

For both ER-stagRFP and ER-mNeonGreen imaging, the image plane laser intensity is maintained at 200 W/cm². The DMD and sCMOS camera are synchronized such that a new speckle illumination is projected only after the current frame is read out on the sCMOS to prevent distortion in images. For a 512 × 512 raw video at 100 fps, the readout time per frame is 5 ms, and the actual exposure time for fluorescence detection is 5 ms.

### Reporting summary

Further information on research design is available in the Nature Portfolio Reporting Summary linked to this article.

## Data availability

The simulated datasets used for training and testing the models presented in this work, along with the experimental data used in the figures, are available on Zenodo at https://zenodo.org/records/17852915.

## Code availability

The code for UBSIM is available on GitHub at https://github.com/Zach-T-Burns/Unrolled-blind-SIM. MATLAB code for generating the simulated datasets is available on Zenodo at https://zenodo.org/records/17852915.

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

## Acknowledgements

This work was supported by the National Science Foundation (CBET-2348536 to Z.L.) and the National Institutes of Health (R35 CA197622 to J.Zhang). Z.B. was supported by the NSF graduate research fellowship program (DGE-2038238 to Z.B.), and A.Z.S. was supported by a fellowship from the National Institute of Dental and Craniofacial Research (1F31DE032886-01A1 to A.Z.S.).

## Author contributions

Z.B. conceived the study, designed the code, and trained the models. Z.B., J.Zhao., and A.Z.S. designed and conducted the experiments. Z.B. processed experimental and simulated data. Z.B. prepared the figures. Z.B., J.Zhao, and A.Z.S. wrote the manuscript. J.Zhang and Z.L. supervised the work and contributed to the discussion.

## Competing interests

The authors declare no competing interests.
