## [Transparent Peer Review file · Nature Communications]

High-speed blind structured illumination microscopy via unsupervised algorithm unrolling

Corresponding Author: Professor Zhaowei Liu

Version 0:

Reviewer comments:

Reviewer #1

(Remarks to the Author)

The manuscript presents an unrolled version of the Blind-SIM (UBSIM) algorithm aimed at reducing inference time. The results are clearly described, and the authors have commendably shared the code and the training dataset, which enhances reproducibility and transparency.

However, I have several concerns and suggestions for improvement:

1. **Novelty Claim:** The authors assert that their approach is the first to achieve video-rate blind-SIM imaging at 50 Hz. While this is an impressive claim, it may be overstated given that other algorithms have been developed to address the same problem (e.g., ref [1] below). A more thorough discussion of related work is needed to contextualize this contribution.
2. **Dataset Size and Reconstruction Validity:** The use of only 20 images for reconstruction seems inadequate for ensuring the validity of equation (2). While this number might suffice for sparse samples, it is likely insufficient for denser samples, where the signal complexity increases. A discussion on the limitations of this approach for varying sample densities would strengthen the manuscript.
3. **Temporal Smoothing:** The authors employ a shift of only two images between consecutive reconstructions, resulting in 18 images being common across adjacent reconstructions. This overlap introduces a temporal smoothing effect, which may artificially enhance the perception of temporal resolution. The implications of this overlap should be explicitly discussed, as it could undermine the claim of achieving true video-rate performance.
4. **Training Requirements:** While unrolled algorithms are known to generalize better and require less training data, the need for training is still a limitation. This aspect is understated in the manuscript and should be more thoroughly addressed to provide a balanced perspective on the advantages and limitations of the proposed approach.

In summary, while the proposed UBSIM algorithm represents a relevant and promising contribution, several claims and methodological choices require further justification or discussion. Addressing these concerns would enhance the manuscript's rigor and provide a clearer understanding of the algorithm's true capabilities and limitations.

[1] Giroussens, Guillaume, et al. "Fast super-resolved reconstructions in fluorescence random illumination microscopy (RIM)." *IEEE Transactions on Computational Imaging* (2024).

(Remarks on code availability)

Reviewer #2

(Remarks to the Author)

see attached file

(Remarks on code availability)

I had a quick overview of the code

Version 1:

Reviewer comments:

Reviewer #2

(Remarks to the Author)

Despite my initially somewhat negative opinion of this work during the first round of review, I must admit that the substantial revisions made by the authors, along with their detailed responses, have led me to change my mind. I would like to congratulate the authors on this very thorough revision, in which they have effectively addressed all of my concerns. For these reasons, I recommend that the paper be accepted for publication.

(Remarks on code availability)

I carefully examined the code during the first round of review. Although I did not recheck it in detail after the revision, the newly reported figures give me confidence that the changes are sound and are indeed consistent with the authors' responses.

Reviewer #1

1. **Comment:** “Novelty Claim: The authors assert that their approach is the first to achieve video-rate blind-SIM imaging at 50 Hz. While this is an impressive claim, it may be overstated given that other algorithms have been developed to address the same problem (e.g., ref [1] below). A more thorough discussion of related work is needed to contextualize this contribution.”

Response:

We thank the reviewer for their comment and recognize the importance of making accurate and nuanced novelty claims for this work. We would like to first note that the specific claim of “video-rate blind-SIM imaging at 50 Hz” depends on the experimental configurations for high-speed data acquisition, while the algorithm provides real time super-resolution display.

There are two notions of speed in regard to super-resolution imaging. One is the **acquisition speed** which describes how long it takes to collect the frames for the imaging system and is limited by hardware. The other is **reconstruction speed** which refers to how fast a super-resolved image can be reconstructed from raw data via an algorithm.

Our work has novelty in regard to *both* notions of speed. The unrolled algorithm method enables exceptional reconstruction speed that is orders of magnitude faster than other algorithms for random speckle imaging. Additionally, we were able to achieve high image acquisition speeds using a DMD to create speckle patterns that are synchronized with a sCMOS camera. This type of reconstruction speed, combined with experimental live-cell video data, has not been demonstrated for blind-SIM imaging.

However, as we recognize there have been other traditional SIM imaging implementations with high capture speed [Ref. 1], we removed the wording “for the first time” from the abstract with regard to capture speed to avoid confusion.

We have modified the main text to clarify the paper’s contributions in terms of both acquisition and reconstruction speed.

[Revision in the main text]

We present unrolled blind-SIM (UBSIM), an algorithm which integrates a learnable neural network inside the unrolled iterations of the blind-SIM algorithm. **UBSIM delivers a reconstruction speed two to three orders of magnitude faster than that of current iterative**

blind-SIM methods, while achieving similar resolution and image quality. Furthermore, we demonstrate that UBSIM can be trained in an unsupervised manner that reduces hallucinations and produces superior generalization capability when compared to benchmark super-resolution networks. We test UBSIM experimentally on live cells and present video-rate super-resolution imaging up to 50 Hz. Using our method, we observe dynamic remodeling of the endoplasmic reticulum with high spatiotemporal resolution.

[Revision in the main text]

Structured illumination microscopy (SIM) is a widely used super-resolution (SR) technique that illuminates a sample with spatially non-uniform light to recover high-resolution information. While SIM has a moderate two-fold resolution improvement when compared to other SR techniques such as STORM or STED, it requires only a handful of raw images to generate a SR frame, and thus provides high temporal resolution from both the data acquisition perspective and also SR reconstruction perspective.

[Revision in the main text]

Several other iterative blind-SIM algorithms have since been developed based on other various constraints. A thorough comparison of our work with the existing blind-SIM literature is provided in the supplementary material (Supp. Table 1).

[Revision in the main text]

In this paper, we propose a new unrolled algorithm we name Unrolled Blind-SIM (UBSIM). UBSIM produces a two-fold improvement in resolution with a reconstruction speed that is two to three orders of magnitude faster than iterative blind-SIM in both simulations and experiments. We demonstrate that UBSIM generalizes to unseen data far better than reference SR DNNs. Lastly, to demonstrate its applicability in biological research, we perform live cell imaging of endoplasmic reticulum dynamics in live cells with a high-speed SIM acquisition setup utilizing a synchronized DMD illumination system. Using UBSIM, we observe the movement and collapse of endoplasmic reticulum network structures in real time.

The supplementary material is also modified to include a table comparing our paper to other blind-SIM papers in the literature:

[Revision in the supplementary material]

S1: Comparison with current blind-SIM literature

Title	Method	Live Cell data?	Video data?	Limitations
Structured illumination microscopy using unknown speckle patterns ¹ (2012)	Iterative physics-based model	No (fixed tissue only)	No	 • Slow reconstruction even with conjugate gradient acceleration • No dynamic, or live cell imaging demonstrated
Structured illumination fluorescence microscopy with distorted excitations using a filtered blind-SIM algorithm ² (2013)	Iterative physics-based model	No (fixed cell only)	No	 • Slow iterative reconstruction method • No live cell or video data
Fluorescent microscopy beyond diffraction limits using speckle illumination and joint support recovery ³ (2013)	Iterative physics-based model	No (fixed cell only)	No	 • Slow iterative reconstruction method • No live cell or video data
Structured illumination microscopy with unknown patterns and a statistical prior ⁴ (2017)	Iterative physics-based model	No	No	 • Slow iterative reconstruction method • No video or live cell demonstrations
Joint Reconstruction Strategy for Structured Illumination Microscopy With Unknown Illuminations ⁵ (2017)	Iterative physics-based model	No	No	 • Requires regularization parameter tuning • Accelerated iterative method still requires more iterations than UBSIM • No video or live cell demonstrations
On the Superresolution Capacity of Imagers Using Unknown Speckle Illuminations ⁶ (2018)	Iterative physics-based model	No	No	 • Purely theoretical paper • Slow iterative reconstruction method • No video or live cell demonstrations
Super-resolved live-cell imaging using random illumination microscopy ⁷ (2021)	Iterative physics-based model	Yes	Yes	 • Slow iterative reconstruction

Deep learning for blind structured illumination microscopy ⁸ (2022)	Deep learning	Yes	No	 • Limited experimental evaluation • No rigorous exploration of generalizability • Requires supervised training scheme
Fast super-resolved reconstructions in fluorescence random illumination microscopy (RIM) ⁹ (2024)	Iterative physics-based model	No	No	 • Accelerated iterative method • Resolution improvement cannot reach 2x in non-noiseless demonstrations • Limited quantitative resolution evaluation • No experimental live cell data

Supplementary Table 1: Summary of current blind-SIM papers and their limitations compared to this work.

Additionally, we thoroughly read the specific reference provided by the reviewer (Giroussens, Guillaume, et al. "Fast super-resolved reconstructions in fluorescence random illumination microscopy (RIM)." IEEE Transactions on Computational Imaging (2024).) [Ref. 2] to assess whether this work effects any of our paper’s claims of novelty. While the work does introduce an accelerated algorithm for the related method of variance-based Random Illumination Microscopy, **we find that the paper does not affect the novelty of our paper or improve upon our UBSIM algorithm for the following reasons:**

- [1] In terms of algorithmic reconstruction speed, the paper introduces a non-iterative reconstruction method termed RIM-CF. However, the authors explain that this method, while better than Wiener deconvolution in some cases, cannot reach the full 2x resolution improvement and that iterative methods are needed for full resolution doubling. See comments from the text such as “*Obviously, even in a favorable scenario, RIM-CF remains based on the approximation (12) and modeling errors deteriorate the resolution gain, e.g., RIM-CF cannot reach the theoretical resolution limit one can expect from RIM in a noise-free setting. An iterative solver, based on a more accurate model is required to reach this limit.*”, “*RIM-CF is basically a non-iterative regularized deconvolution*” (section III. A non-iterative estimator, RIM-CF), and “*RIM-CF is a fast, FFT-based inversion of the standard deviation that provides a super-resolved estimate of the sample. However, because it is based on an approximation, this estimator cannot achieve the maximal theoretical resolution gain for RIM.*” (Conclusion) Furthermore, this lack of resolution gain from the non-

iterative method can be clearly seen in Figure 2. Here, **even in a noiseless demonstration case**, RIM-CF is unable to reach full resolution improvement compared to RIM-STD/RIM-VAR.

- [2] The paper also introduces an iterative algorithm RIM-STD which is an accelerated version of the earlier developed RIM-VAR algorithm. However, there are several aspects in which this algorithm does not match our paper's performance. Firstly, the number of iterations needed for convergence in the noiseless 2D imaging case (Figure 2) is 100 iterations which is still an order of magnitude more than our UBSIM algorithm. Secondly, it appears that RIM-STD only can reach a full 2x resolution improvement in highly idealized, simulated scenarios. In Figure 3, RIM-STD achieves the full resolution improvement under the "asymptotic variance" case where essentially the variance is perfectly known. However, when the empirical variance must be calculated, which is what would happen in real-world imaging, the performance of RIM-STD is much worse. Even with 1,000 sub-images and a very high SNR (30 dB) RIM-STD is unable to achieve the full resolution improvement provided by the speckle and in fact it is stated that "*RIM-CF and RIM-STD now reach comparable results in terms of quality*". That means in the experimental case, RIM-STD does not perform any better than a form of regularized deconvolution and cannot reach full resolution improvement.
- [3] We find the paper has several issues in how resolution is determined and presented. The only figures for which quantitative resolution improvement or intensity cross sections are provided for are Figures 1 and 2 which are both simulated target objects with *noiseless* data collection. However, this is problematic for two reasons: (1) in the case of zero noise (infinite SNR) super-resolution is somewhat trivial. Using the known PSF, the super-resolution object can be perfectly recovered via bandwidth extrapolation for many synthetic objects [see Refs. 3,4,5], and (2) for Figure 1 a single point source is used as the resolution measurement object. This again is a trivial problem as there exist many algorithms that can arbitrarily sharpen a single point source and thus does not provide an accurate measure of resolution. In microscopy, resolution is defined as the distance at which two closely spaced point sources can be distinguished (Abbe, Rayleigh) and not the FWHM of a single point source. For example, simple Lucy-Richardson deconvolution can arbitrarily sharpen a single point's width in a similar manner with few iterations, however it is not a true super-resolution method as it will fail to resolve closely spaced point sources in an environment with realistic noise levels.
- [4] Among the provided figures, only figure 4 contains reconstruction on experimental data. The data is of sparse fixed fluorescent beads and there is no reconstruction on fixed or live cell data that could provide dynamic behavior.
- [5] We find that the paper does not contain any video reconstructions, only single frame reconstructions (see figures 1- 4, no supplementary available).

To conclude, the paper “Fast super-resolved reconstructions in fluorescence random illumination microscopy (RIM)” is not comparable to our paper as it presents (1) a non-iterative algorithm that is only slightly better than deconvolution and cannot reach full resolution improvement even in noiseless conditions, and (2) an accelerated iterative algorithm that improves upon RIM-VAR, but cannot reach full resolution improvement in any realistic scenario with noise and is still 1 order of magnitude slower than our algorithm. The paper mainly focuses on highly idealized simulations and does not provide live-cell or video imaging.

2. **Comment:** “Dataset Size and Reconstruction Validity: The use of only 20 images for reconstruction seems inadequate for ensuring the validity of equation (2). While this number might suffice for sparse samples, it is likely insufficient for denser samples, where the signal complexity increases. A discussion on the limitations of this approach for varying sample densities would strengthen the manuscript.”

Response:

We thank the reviewer for their comment and recognize that importance of clarifying for the reader the relationship between number of sub-frames and image reconstruction performance in blind-SIM imaging.

For fully developed speckle the contrast of L summed frames will be reduced at a rate of $\frac{1}{\sqrt{L}}$ (see J. W. Goodman, “Some fundamental properties of speckle,” Journal of the Optical Society of America (1976). [Ref. 6]). We can derive an expression for the RMS error by doing a bias and variance analysis of the uniform intensity estimator. Starting from:

$$\sum_{l=1}^L I_l \approx LI_0$$

We define the fractional error as:

$$E_l = \frac{\sum_{l=1}^L I_l - LI_0}{LI_0}$$

For fully developed speckle, the intensity follows an independent and identically distributed exponential random variable with mean I_0 and variance I_0^2 .

The expected value of the error (bias) is then:

$$\mathbb{E}(E_l) = \frac{\mathbb{E}(\sum_{l=1}^L I_l) - LI_0}{LI_0} = \frac{\sum_{l=1}^L \mathbb{E}(I_l) - LI_0}{LI_0} = \frac{LI_0 - LI_0}{LI_0} = 0$$

And the variance of the error is:

$$\text{Var}(E_l) = \text{Var}\left(\frac{\sum_{l=1}^L I_l - LI_0}{LI_0}\right) = \text{Var}\left(\frac{\sum_{l=1}^L I_l}{LI_0} - 1\right) = \frac{\text{Var}(\sum_{l=1}^L I_l)}{(LI_0)^2} = \frac{LI_0^2}{L^2 I_0^2} = \frac{1}{L}$$

Mean squared error can then be calculated as the decomposition:

$$\text{MSE}(E_l) = \text{Bias}(E_l)^2 + \text{Var}(E_l) = \frac{1}{L}$$

Giving a RMSE of:

$$\text{RMSE} = \frac{1}{\sqrt{L}}$$

Thus, the residual error in the uniformity constraint will decrease with an inverse square relationship with the number of added frames.

Practically, the result of this error will lead to (1) a multiplicative shading of the resulting image and (2) higher noise which will result in artifacts [Refs. 7,8]. The resolution, however, is mainly determined by the OTF and the spatial frequencies of the illumination patterns.

The results of smaller frame number can be seen in Mudry et al.'s original paper [Ref. 7] (Supplementary Figure 4) where a smaller number of frames are used for reconstruction of the Siemen star object than the main text (note that the Siemen star is a relatively dense object). The resulting reconstruction still has the same resolution as periodic SIM with 9 known frames. However, there is additional granularity introduced by multiplicative shading and noise.

Ultimately, the number of frames is a parameter that must be determined by the user with diminishing returns as the number of frames increases. There are other blind-SIM papers which have demonstrated reconstruction in the range of 20-40 frames [Refs. 8,9].

We clarified the relationship between number of frames and uniformity error to make it clearer to the reader. Additionally, we added Supplementary Section S3 which includes the error derivation shown here along with a discussion on the effects on reconstruction results.

[Revision in the supplementary material]

In this section, an error analysis for the uniformity assumption of the blind-SIM algorithm is presented. The bias, variance, and root mean square error of the estimator is derived as a function of the number of sub-frames used. We also discuss how error in the uniformity assumption affects the reconstructed image.

The uniformity assumption is defined as:

$$\sum_{l=1}^L I_l \approx LI_0$$

The assumption is that sum of all illumination patterns is the same as the average intensity multiplied by the number of frames.

The fractional error is then defined as:

$$E_l = \frac{\sum_{l=1}^L I_l - LI_0}{LI_0}$$

For fully developed speckle¹⁰, the intensity follows an independent and identically distributed (i.i.d) exponential random variable with mean I_0 and variance I_0^2 .

The expected value of the error (bias of the estimator) is then:

$$\mathbb{E}(E_l) = \frac{\mathbb{E}(\sum_{l=1}^L I_l) - LI_0}{LI_0} = \frac{\sum_{l=1}^L \mathbb{E}(I_l) - LI_0}{LI_0} = \frac{LI_0 - LI_0}{LI_0} = 0$$

The variance of the estimator is:

$$\text{Var}(E_l) = \text{Var}\left(\frac{\sum_{l=1}^L I_l - LI_0}{LI_0}\right) = \text{Var}\left(\frac{\sum_{l=1}^L I_l}{LI_0} - 1\right) = \frac{\text{Var}(\sum_{l=1}^L I_l)}{(LI_0)^2} = \frac{LI_0^2}{L^2 I_0^2} = \frac{1}{L}$$

Mean squared error can then be calculated from the decomposition:

$$\text{MSE}(E_l) = \text{Bias}(E_l)^2 + \text{Var}(E_l) = \frac{1}{L}$$

Giving a root mean squared error of:

$$RMSE(E_l) = \frac{1}{\sqrt{L}}$$

Therefore, the residual error in the uniformity constraint will decrease with an inverse square relationship with the number of sub-frames used for reconstruction. Adding more frames will always result in reduced error, however at a diminishing rate of return.

The presence of error in the uniformity constraint will lead to (1) a multiplicative shading of the resulting image and (2) higher noise which will result in granular artifacts^{1,4}. The resolution of reconstructed image is ultimately determined by the spatial frequency support of the optical transfer function (OTF) and the illumination speckle. However, artifacts from shading and lowered signal to noise ratio from fewer sub-frames may reduce apparent resolution. Ultimately, the number of sub-frames used is a parameter to be chosen by the user that will cause a tradeoff between shading accuracy and temporal resolution.

[Revision in the main text]

This is achieved by making an additional assumption that all the illumination patterns on average sum to a uniform field:

$$\sum_{l=1}^L I_l \approx LI_0$$

Where L is the number of sub-frames and I_0 is a constant equal to the average intensity received by the sample. The error in the uniform approximation drops at a rate of $\frac{1}{\sqrt{L}}$. See Supplementary Section S3 for derivation.

3. **Comment:** “Temporal Smoothing: The authors employ a shift of only two images between consecutive reconstructions, resulting in 18 images being common across adjacent reconstructions. This overlap introduces a temporal smoothing effect, which may artificially enhance the perception of temporal resolution. The implications of this overlap should be explicitly discussed, as it could undermine the claim of achieving true video-rate performance.”

Response:

We thank the reviewer for their comment and agree that the sliding-frame window can introduce temporal smoothing and that it is important to make this point clear to the reader. Our approach can introduce a smoothing effect but can be appropriate when the sample movement is generally not faster than the SR frame integration time [Refs. 10,11]. We included a paragraph in discussion section that mentions the limitations of such an approach.

[Revision in the main text]

We modulated the illumination patterns at 20 fps and used 20 frames for reconstruction. To visualize dynamics, we used a rolling-frame reconstruction method with a 2-frame step to achieve an effective reconstructed video speed of 10 fps. Because this method includes a large overlap in frames between reconstructions, it introduces temporal smoothing, but can be appropriate for ER where the dynamics of interest (compartment remodeling) occur on timescales of seconds and are generally slower than the total SR frame integration time^{57, 58}.

[Revision in the main text]

Secondly, UBSIM, like all blind-SIM methods, requires multiple sub-frames for each reconstructed SR frame which can affect image acquisition speeds. In our work we utilize a sliding-frame reconstruction approach in addition to our high-speed illumination system to boost acquisition speed. However, this approach can cause temporal smoothing and is not suitable for cases where the sample movement is generally faster than the effective integration time for a single SR frame. Further improvements in blind-SIM acquisition speed require either a reduction in sub-frames per reconstructed image or increased speckle modulation and camera capture rates.

4. **Comment:** “Training Requirements: While unrolled algorithms are known to generalize better and require less training data, the need for training is still a limitation. This aspect is understated in the manuscript and should be more thoroughly addressed to provide a balanced perspective on the advantages and limitations of the proposed approach.”

Response:

We thank the reviewer for their comment and agree that the need for training is a limitation of the method. There is a tradeoff between the upfront time needed to train the model versus the time savings provided by UBSIM that should be considered by a potential user. For one-time reconstructions of a single image, it would make sense to use an iterative reconstruction method as the longer inference time is still much shorter than training the model. However,

for applications where a user will be doing many repeated image reconstructions, such as taking videos, or fixed setups where repeated experiments are being run, UBSIM can provide a significant advantage. Additionally, compared to other learning-based reconstruction methods, UBSIM does not need to be retrained depending on the object type due to its strong generalization capability.

A likely scenario for deciding between reconstruction methods would be using iterative methods for initial testing of a setup and parameters and switching to UBSIM for repeated/video imaging afterwards.

[Revision in the main text]

While UBSIM provides improvements over current blind-SIM algorithms, there are several limitations that must be considered by a potential user. Firstly, UBSIM requires training before inference use, which creates an upfront time cost. For one-off reconstructions, iterative methods, even with their longer reconstruction times, may be preferred. However, for cases that include repeated image reconstruction, such as video imaging or repeated experiments, UBSIM can provide a significant time advantage.

Reviewer #2

1. **Comment:** “The constraint (2) is used to obtain a closed system with L unknowns and L equations. However, how is I_0 determined? The value assigned to it will influence the reconstructed image.”

Response:

We thank the author for the comment and realize this can be a source of confusion. We set I_0 the same way as described in the original paper from Mudry et. al. [Ref. 7] “ I_0 is a constant equal to the average intensity received by the sample” (see supplementary material for “Structured Illumination microscopy using unknown speckle patterns”, section 1. Blind-SIM and deconvolution algorithms). In order to make this clearer for the reader, we added additional sentence to the manuscript in the section “Derivation of unrolled algorithm.”

[Revision in the main text]

This is achieved by making an additional assumption that all the illumination patterns on average sum to a uniform field:

$$\sum_{l=1}^L I_l \approx LI_0 \quad (2)$$

Where L is the number of sub-frames and I_0 is a constant equal to the average intensity received by the sample

In our code, I_0 is estimated by the following procedure. At each pixel, the average recorded signal over L frames is the product of the average illumination intensity and fluorophore density: $S(x) = I_0\rho(x)$. Thus, the effective I_0 , after normalization, can be found by averaging the input frames tensor along the frame dimension and taking the maximum of the per-pixel average intensity. This can be seen in the provided code in the initialization of variable I_0 in `Iterative_Blind_SIM()` and `Unrolled_Blind_SIM()`.

2. **Comment:** “In (4), the authors present a gradient descent algorithm to minimize the data fidelity term (3) (without any regularization). Then, in (5), they propose an unrolled version by inserting a CNN around the gradient step. The authors interpret this CNN as a proximal

operator. However, this interpretation is unusual and not clearly justified. Typically, in algorithm unrolling where a network is used in place of a prox, one starts from a well-established proximal algorithm and replaces each proximal step with a learned CNN. In (5), it is unclear which underlying algorithm is being unrolled. If the CNN is meant to represent a proximal operator, the resulting structure does not correspond to any standard algorithm I am aware of. A common and well-understood strategy is to unroll a proximal gradient algorithm, where the CNN is applied around the entire gradient descent step, thus acting as a learned prior or regularization term. In the current setup, given how the CNN is positioned, it would be more natural to interpret it as a preconditioner rather than a proximal operator. This distinction needs to be clarified by the authors.”

Response:

We thank the reviewer for their comment and agree with their interpretation of our method. It is true that our approach is different from many unrolled algorithm implementations as in our method the CNN operates directly on the gradient instead of operating on the object after a gradient step has already been applied as a proximal step does. This indeed is more similar to gradient preconditioning. This approach could also be interpreted as a form of learned gradient descent where the update direction function is learned. The CNN operates on the gradient to calculate an update direction similar to how the conjugate gradient method calculates an update direction using the gradient. We agree with the author that our use of proximal operator was incorrect and that we should clarify as well as provide relevant references for these interpretations.

[Revision in the main text]

The gradients (stacked as channels in a tensor object) are then used as inputs to a CNN. The scaled outputs of the CNN are added to the original inputs to get the updated object and illumination patterns. **The CNN operates directly on the gradients similar to how a preconditioner^{32, 33} or update direction calculation^{31,34,35} is used to accelerate convergence for iterative gradient descent algorithms. In this way, the use of the network can be thought of as a type of learned preconditioner^{36, 37} or learned gradient descent^{38,39,40,41} for which there are examples from literature.**

3. **Comment:** “The iterative algorithm considered is a simple gradient descent, which is well known to converge slowly, with a rate of $O(1/k)$. However, it would be straightforward to implement an accelerated version using Nesterov's method, requiring only minor modifications to the code and achieving a faster convergence rate of $O(1/k^2)$. Based on my experience with imaging inverse problems, such an acceleration significantly improves convergence speed. I believe that Fig. 3 should be complemented with a comparison

including Nesterov acceleration, in order to better highlight the relevance and advantages of the proposed unrolled method.”

Response:

We thank the reviewer for their comment and agree that including a comparison to an accelerated method such as gradient descent with Nesterov momentum strengthens the paper and provides a stronger baseline to compare our unrolled method against. We have revised Figure 3 of our manuscript to include results using gradient descent with Nesterov momentum. Implementation of the method can be viewed in our attached code as well.

Our results show that the Nesterov method reduces both the average iterations and computation time by approximately one order of magnitude over plain gradient descent. However, UBSIM still requires one order of magnitude less iterations and is two orders of magnitude faster in computation time compared to the Nesterov method. Thus, our unrolled method still displays a strong speed improvement over accelerated iterative methods.

[Revision in the main text]

The Polak-Ribière conjugate gradient method³¹, used by Mudry and colleagues, as well as Nesterov acceleration³², can also be used to significantly accelerate optimization. Details of the implementation of these methods are included in Supplementary Section S4.

[Revision in the main text]

Fig 3: Inference time comparison of iterative methods versus unrolled blind-SIM. a, Comparison of computation time for image reconstruction. **b,** Comparison of normalized loss versus iteration number. Lines are the average values for 100 test set images. Dashed black line marks the average normalized loss for the unrolled network after 6 iterations.

[Revision in the main text]

In Fig. 3 we quantitatively compared the speed for unrolled and iterative blind-SIM. For iterative blind-SIM, we ran evaluations using regular gradient descent as well as two accelerated methods: nonlinear conjugate gradient descent³¹ and gradient descent with Nesterov momentum³². Additional information on the implementation of these methods is included in Supplementary Section S4.

[Revision in the main text]

We found that UBSIM was on average 3 orders of magnitude faster than the gradient descent implementation and 2 orders of magnitude faster than the conjugate gradient descent and Nesterov acceleration methods in terms of computation time. UBSIM required 2 orders of magnitude fewer iterations than gradient descent and 1 order fewer iterations than the accelerated methods. A single 256 x 256 pixel image took approximately 10-20 seconds to reconstruct for gradient descent, 1-3 seconds for the accelerated methods, and 10 milliseconds for UBSIM.

4. **Comment:** “In the supplementary section 5, the authors show that conjugate gradient is not faster than a simple gradient descent, which is very surprising to me. How is the step size α_{\pm} chosen? From the provided code I can see that the value of α_{\pm} is fixed by hand and reduced along with iteration in a very heuristic manner... In contrast, in Mudry et al. (and in general in conjugate gradient), it is optimized at each iteration. Also what about the non-negativity constraint imposed (through squaring the relevant variables) in Mudry et al.?”

Response:

We thank the reviewer for their comment and agree that for a fair comparison with the nonlinear conjugate gradient method a more rigorous line search method is needed. In our original implementation we used a simple backtracking line search that dropped step size by $\frac{1}{2}$ when the cost function did not decrease. This did not yield a significant difference from gradient descent. In order to achieve the true acceleration of the conjugate gradient method we implemented an inexact line search method that optimizes at each step for strong Wolfe conditions. The strong Wolfe conditions ensure a (1) sufficient decrease in the cost function and (2) strong curvature of the descent direction. [Ref. 12]. Details on the new implementation of the conjugate gradient method and line search are provided in

Supplementary section S4. Additionally, the provided code has been updated to include the new line search method.

The results for the new line search method are shown in the updated Figure 3 along with gradient descent, Nesterov acceleration, and UBSIM. The conjugate gradient method provides an approximately 1 order reduction in number of iterations and a 1 order improvement in compute speed. The conjugate gradient method provides a similar speed improvement as the Nesterov accelerated implementation. The conjugate gradient method requires less iterations, although is slightly longer in compute time on average due to the more complex optimization at each step.

The results show that UBSIM is still 1 order faster than the conjugate gradient method in terms of iterations and 2 orders faster in terms of compute time.

[Revision in the main text]

The Polak-Ribière conjugate gradient method³¹, used by Mudry and colleagues, as well as Nesterov acceleration³², can also be used to significantly accelerate optimization. Details of the implementation of these methods are included in Supplementary Section S4.

[Revision in the main text]

Fig 3: Inference time comparison of iterative methods versus unrolled blind-SIM. **a**, Comparison of computation time for image reconstruction. **b**, Comparison of normalized loss versus iteration number. Lines are the average values for 100 test set images. Dashed black line marks the average normalized loss for the unrolled network after 6 iterations.

[Revision in the main text]

In Fig. 3 we quantitatively compared the speed for unrolled and iterative blind-SIM. For iterative blind-SIM, we ran evaluations using regular gradient descent as well as two accelerated methods: nonlinear conjugate gradient descent³¹ and gradient descent with Nesterov momentum³⁴. Additional information on the implementation of these methods is included in Supplementary Section S4.

[Revision in the main text]

We found that UBSIM was on average 3 orders of magnitude faster than the gradient descent implementation and 2 orders of magnitude faster than the conjugate gradient descent and Nesterov acceleration methods in terms of computation time. UBSIM required 2 orders of magnitude fewer iterations than gradient descent and 1 order fewer iterations than the accelerated methods. A single 256 x 256 pixel image took approximately 10-20 seconds to reconstruct for gradient descent, 1-3 seconds for the accelerated methods, and 10 milliseconds for UBSIM.

In regard to the reviewer’s concern over the non-negativity constraint (Also what about the non-negativity constraint imposed (through squaring the relevant variables) in Mudry et al.?) we would like to assure them that in our original provided code we do implement the non-negativity constraint through variable squaring. We realize there may have been confusion based on our description in the section “Derivation of unrolled algorithm” where we do not mention this step. To make this clearer, we have added text to this section explaining how variable squaring is used to ensure non-negativity.

[Revision in the main text]

To ensure positivity, I_l and ρ are rewritten as the square of auxiliary variables such that:

$$\begin{aligned} I_l &= i_l^2 \\ \rho &= \xi^2 \end{aligned} \tag{3}$$

5. **Comment:** “The blind SIM iterative algorithm proposed by Mudry et al. is now more than 10 years old. Since then, faster non-iterative alternatives have been proposed such as [1] Giroussens et al (2024). Fast super-resolved reconstructions in fluorescence random illumination microscopy (RIM). IEEE Transactions on Computational Imaging.”

Response:

We thank the reviewer for their comment and agree that it is important to compare our work with other blind-SIM papers that have been published since Mudry et al. We have added Supplementary Section S1 (which is referenced in the introduction section) to compare our work to other blind-SIM papers that have been published in terms of speed, method, and types of experimental data used. From our findings UBSIM is state of the art in terms of reconstruction speed. In addition, our work is the only paper to demonstrate high-speed reconstruction along with video live cell imaging while most other blind-SIM papers have only shown single-frame bead or fixed cell reconstructions.

[Revision in the main text]

Several other iterative blind-SIM algorithms have since been developed based on other various constraints. **A thorough comparison of our work with the existing blind-SIM literature is provided in the supplementary material (Supp. Table 1).**

[Revision in the supplementary material]**S1: Comparison with current blind-SIM literature**

Title	Method	Live Cell data?	Video data?	Limitations
Structured illumination microscopy using unknown speckle patterns ¹ (2012)	Iterative physics-based model	No (fixed tissue only)	No	 • Slow reconstruction even with conjugate gradient acceleration • No dynamic, or live cell imaging demonstrated
Structured illumination fluorescence microscopy with distorted excitations using a filtered blind-SIM algorithm ² (2013)	Iterative physics-based model	No (fixed cell only)	No	 • Slow iterative reconstruction method • No live cell or video data
Fluorescent microscopy beyond diffraction limits using speckle illumination and joint support recovery ³ (2013)	Iterative physics-based mode	No (fixed cell only)	No	 • Slow iterative reconstruction method • No live cell or video data

Structured illumination microscopy with unknown patterns and a statistical prior ⁴ (2017)	Iterative physics-based model	No	No	 • Slow iterative reconstruction method • No video or live cell demonstrations
Joint Reconstruction Strategy for Structured Illumination Microscopy With Unknown Illuminations ⁵ (2017)	Iterative physics-based model	No	No	 • Requires regularization parameter tuning • Accelerated iterative method still requires more iterations than UBSIM • No video or live cell demonstrations
On the Superresolution Capacity of Imagers Using Unknown Speckle Illuminations ⁶ (2018)	Iterative physics-based model	No	No	 • Purely theoretical paper • Slow iterative reconstruction method • No video or live cell demonstrations
Super-resolved live-cell imaging using random illumination microscopy ⁷ (2021)	Iterative physics-based model	Yes	Yes	 • Slow iterative reconstruction
Deep learning for blind structured illumination microscopy ⁸ (2022)	Deep learning	Yes	No	 • Limited experimental evaluation • No rigorous exploration of generalizability • Requires supervised training scheme
Fast super-resolved reconstructions in fluorescence random illumination microscopy (RIM) ⁹ (2024)	Iterative physics-based model	No	No	 • Accelerated iterative method • Resolution improvement cannot reach 2x in non-noiseless demonstrations • Limited quantitative resolution evaluation • No experimental live cell data

Supplementary Table 1: Summary of current blind-SIM papers and their limitations compared to this work.

We also would like to specifically address the paper mentioned by the reviewer (Giroussens et al (2024). Fast super-resolved reconstructions in fluorescence random illumination microscopy (RIM). IEEE Transactions on Computational Imaging) [Ref. 2]. We have thoroughly read the paper to assess whether this work effects any of our paper's claims of novelty. While the work does introduce an accelerated algorithm for the related method of variance-based Random Illumination Microscopy, **we find that the paper does not affect the novelty of our paper or improve upon our UBSIM algorithm for the following reasons:**

- [1] In terms of algorithmic reconstruction speed, the paper introduces a non-iterative reconstruction method termed RIM-CF. However, the authors explain that this method, while better than Wiener deconvolution in some cases, cannot reach the full 2x resolution improvement and that iterative methods are needed for full resolution doubling. See comments from the text such as *“Obviously, even in a favorable scenario, RIM-CF remains based on the approximation (12) and modeling errors deteriorate the resolution gain, e.g., RIM-CF cannot reach the theoretical resolution limit one can expect from RIM in a noise-free setting. An iterative solver, based on a more accurate model is required to reach this limit.”*, *“RIM-CF is basically a non-iterative regularized deconvolution”* (section III. A non-iterative estimator, RIM-CF), and *“RIM-CF is a fast, FFT-based inversion of the standard deviation that provides a super-resolved estimate of the sample. However, because it is based on an approximation, this estimator cannot achieve the maximal theoretical resolution gain for RIM.”* (Conclusion) Furthermore, this lack of resolution gain from the non-iterative method can be clearly seen in Figure 2. Here, **even in a noiseless demonstration case**, RIM-CF is unable to reach full resolution improvement compared to RIM-STD/RIM-VAR.
- [2] The paper also introduces an iterative algorithm RIM-STD which is an accelerated version of the earlier developed RIM-VAR algorithm. However, there are several aspects in which this algorithm does not match our paper's performance. Firstly, the number of iterations needed for convergence in the noiseless 2D imaging case (Figure 2) is 100 iterations which is still an order of magnitude more than our UBSIM algorithm. Secondly, it appears that RIM-STD only can reach a full 2x resolution improvement in highly idealized, simulated scenarios. In Figure 3, RIM-STD achieves the full resolution improvement under the “asymptotic variance” case where essentially the variance is perfectly known. However, when the empirical variance must be calculated, which is what would happen in real-world imaging, the performance of RIM-STD is much worse. Even with 1,000 sub-images and a very high SNR (30 dB) RIM-STD is unable to achieve the full resolution improvement provided by the speckle and in fact it is stated that *“RIM-CF and RIM-STD now reach comparable results in terms of quality”*. That means in the experimental case, RIM-

STD does not perform any better than a form of regularized deconvolution and cannot reach full resolution improvement.

- [3] We find the paper has several issues in how resolution is determined and presented. The only figures for which quantitative resolution improvement or intensity cross sections are provided for are Figures 1 and 2 which are both simulated target objects with *noiseless* data collection. However, this is problematic for two reasons: (1) in the case of zero noise (infinite SNR) super-resolution is somewhat trivial. Using the known PSF, the super-resolution object can be perfectly recovered via bandwidth extrapolation for many synthetic objects [see Refs. 3,4,5], and (2) for Figure 1 a single point source is used as the resolution measurement object. This again is a trivial problem as there exist many algorithms that can arbitrarily sharpen a single point source and thus does not provide an accurate measure of resolution. In microscopy, resolution is defined as the distance at which two closely spaced point sources can be distinguished (Abbe, Rayleigh) and not the FWHM of a single point source. For example, simple Lucy-Richardson deconvolution can arbitrarily sharpen a single point's width in a similar manner with few iterations, however it is not a true super-resolution method as it will fail to resolve closely spaced point sources in an environment with realistic noise levels.
- [4] Among the provided figures, only figure 4 contains reconstruction on experimental data. The data is of sparse fixed fluorescent beads and there is no reconstruction on fixed or live cell data that could provide dynamic behavior.
- [5] We find that the paper does not contain any video reconstructions, only single frame reconstructions (see figures 1- 4, no supplementary available).

To conclude, the paper “Fast super-resolved reconstructions in fluorescence random illumination microscopy (RIM)” is not comparable to our paper as it presents (1) a non-iterative algorithm that is only slightly better than deconvolution and cannot reach full resolution improvement even in noiseless conditions, and (2) an accelerated iterative algorithm that improves upon RIM-VAR, but cannot reach full resolution improvement in any realistic scenario with noise and is still 1 order of magnitude slower than our algorithm. The paper mainly focuses on highly idealized simulations and does not provide live-cell or video imaging.

6. **Comment:** “Widespread adaption and use of SR technologies by target groups” , → adoption ?”

Response:

We agree and have changed the typo.

7. **Comment:** “In Supp file, after Fig S6: "We examined the reconstruction results of UBSIM and iterative blind-SIM when there is a low signal-to-noise ratio. As can be seen in Supplementary Figure 5" → It is Supplementary Figure 6.”

Response:

We have modified the text to ensure that it corresponds to the correct supplementary figure and to follow the correct numbering convention for figures in the supplementary material. There has been a change to the order since revision as well.

8. **Comment:** “Page 14 line 301: "On axis elements of the matrix" → On diagonal elements ?”

Response:

We agree and have changed the text.

9. **Comment:** “Page 17, line 350: "... make it a particularly challenging organelle to image" → the word organelle seems misplaced”

Response:

Endoplasmic reticulum is considered an organelle. It is referenced as such in biology textbooks (see: The Cell: A Molecular Approach [Ref. 13]) as well as the references provided in the manuscript [Refs. 14-17]). An organelle is “A small structure in a cell that is surrounded by a membrane and has a specific function”

(<https://www.cancer.gov/publications/dictionaries/cancer-terms/def/organelle>)

References:

1. Markwirth, A., et al. Video-rate multi-color structured illumination microscopy with simultaneous real-time reconstruction. *Nat. Commun.* **10**, 4315 (2019).
2. Giroussens, G., et al. Fast super-resolved reconstructions in fluorescence random illumination microscopy (RIM). *IEEE Trans. Comput. Imaging*, (2024).
3. Goodman, J. W. *Introduction to Fourier Optics* (Roberts and Company Publishers, 2005).
4. Lindberg, J. Mathematical concepts of optical superresolution. *J. Opt.* **14**, 083001 (2012).
5. Bertero, M. & De Mol, C. *Super-Resolution by Data Inversion*, Vol. 36 (Elsevier, 1996).
6. Goodman, J. W. Some fundamental properties of speckle. *JOSA* **66**, 1145-1150 (1976).
7. Mudry, E. *et al.* Structured illumination microscopy using unknown speckle patterns. *Nat. Photonics* **6**, 312–315 (2012)
8. Yeh, L.-H., Tian, L. & Waller, L. Structured illumination microscopy with unknown patterns and a statistical prior. *Biomed. Opt. Express* **8**, 695–711 (2017).
9. Mangeat, T., et al. Super-resolved live-cell imaging using random illumination microscopy. *Cell Rep. Methods* **1**, (2021).
10. Huang, X., et al. Fast, long-term, super-resolution imaging with Hessian structured illumination microscopy. *Nat. Biotechnol.* **36**, 451-459 (2018).
11. Zhao, W. et al. Sparse deconvolution improves the resolution of live-cell super-resolution fluorescence microscopy. *Nat. Biotechnol.* **40**, 606-617 (2022).
12. Nocedal, J., & Wright, S. *Numerical Optimization*. (Springer, New York, 2006).
13. Cooper, G.M., & Adams, K. *The cell: a molecular approach*. (Oxford Univ. Press, New York, 2022).
14. Schwarz, D. S. & Blower, M. D. The endoplasmic reticulum: structure, function and response to cellular signaling. *Cell. Mol. Life Sci.* **73**, 79–94 (2016)
15. Pain, C. & Kriechbaumer, V. Defining the dance: quantification and classification of endoplasmic reticulum dynamics. *J. Exp. Bot.* **71**, 1757–1762 (2020)
16. Shibata, Y., Voeltz, G. K. & Rapoport, T. A. Rough sheets and smooth tubules. *Cell* **126**, 435–439 (2006)
17. Costantini, L. & Snapp, E. Probing endoplasmic reticulum dynamics using fluorescence imaging and photobleaching techniques. *Curr. Protoc. Cell Biol.* **60**, 21–7 (2013)

Report on: High-speed blind structured illumination microscopy via unsupervised algorithm unrolling

In this work, the authors propose to unroll the Blind SIM algorithm of *Mudry et al.* Numerical experiments show that the proposed unrolled version achieves the same quality in a much lower computational time.

I have several concerns regarding this work (detailed below). First, it appears that the implementation of the original method by *Mudry et al.* is not accurately reproduced. Second, the only benefit of the proposed unrolled method is the improved speed. However, before turning to unrolled networks, there exist numerous standard optimization algorithms that are significantly faster than the simple gradient descent baseline used here (i.e., a properly implemented conjugate gradient method, an (accelerated) gradient descent with a sound line search, or even second-order methods). The true interest of the proposed unrolling approach would be justified only if it demonstrated improvements over these stronger baselines. Lastly, since the publication of *Mudry et al.*, non-iterative methods for speckle blind SIM have been proposed and should also be considered as baselines for a fair comparison.

For all these reasons, I believe that this manuscript is not suitable for publication in Nature Communications.

Comments

- The constraint (2) is used to obtain a closed system with L unknowns and L equations. However, how is I_0 determined? The value assigned to it will influence the reconstructed image.
- In (4), the authors present a gradient descent algorithm to minimize the data fidelity term (3) (without any regularization). Then, in (5), they propose an unrolled version by inserting a CNN around the gradient step. The authors interpret this CNN as a proximal operator. However, this interpretation is unusual and not clearly justified. Typically, in algorithm unrolling where a network is used in place of a prox, one starts from a well-established proximal algorithm and replaces each proximal step with a learned CNN. In (5), it is unclear which underlying algorithm is being unrolled. If the CNN is meant to represent a proximal operator, the resulting structure does not correspond to any standard algorithm I am aware of. A common and well-understood strategy is to unroll a proximal gradient algorithm, where the CNN is applied around the entire gradient descent step, thus acting as a learned prior or regularization term. In the current setup, given how the CNN is positioned, it would be more natural to interpret it as a preconditioner rather than a proximal operator. This distinction needs to be clarified by the authors.
- The iterative algorithm considered is a simple gradient descent, which is well known to converge slowly, with a rate of $O(1/k)$. However, it would be straightforward to implement an accelerated version using Nesterov's method, requiring only minor modifications to the code and achieving a faster convergence rate of $O(1/k^2)$. Based on my experience with imaging inverse problems, such an acceleration significantly improves convergence speed.

I believe that Fig. 3 should be complemented with a comparison including Nesterov acceleration, in order to better highlight the relevance and advantages of the proposed unrolled method.

- In the supplementary section 5, the authors show that conjugate gradient is not faster than a simple gradient descent, which is very surprising to me. How is the step size α chosen? From the provided code I can see that the value of α is fixed by hand and reduced along with iteration in a very heuristic manner... In contrast, in *Mudry et al.* (and in general in conjugate gradient), it is optimized at each iteration. Also what about the non-negativity constraint imposed (through squaring the relevant variables) in *Mudry et al.*?
- The blind SIM iterative algorithm proposed by *Mudry et al.* is now more than 10 years old. Since then, faster non-iterative alternatives have been proposed such as *Giroussens et al (2024). Fast super-resolved reconstructions in fluorescence random illumination microscopy (RIM). IEEE Transactions on Computational Imaging.*

Typos

- "Widespread adaption and use of SR technologies by target groups" → adoption ?
- In Supp file, after Fig S6: "We examined the reconstruction results of UBSIM and iterative blind-SIM when there is a low signal-to-noise ratio. As can be seen in Supplementary Figure 5" → It is Supplementary Figure 6.
- Page 14 line 301: "On axis elements of the matrix" → On diagonal elements ?
- Page 17, line 350: "... make it a particularly challenging organelle to image" → the word organelle seems misplaced